# Effect of a Protestant Work Ethic on Burnout: Mediating Effect of Emotional Dissonance and Moderated Mediating Effect of Negative Emotion Regulation

**Sukbong Choi** [1], **Yungil Kang** [2] **and Kyunghwan Yeo** [3,*]

1 College of Global Business, Korea University, 2511 Sejong-ro, Sejong City 30019, Korea; sukchoi@korea.ac.kr
2 Graduate School of Business, Sakarya University, isletme enstitusu, Esentepe Kampusu Serdivan Sakarya, Sakarya 54050, Turkey; gil.kang@ogr.sakarya.edu.tr
3 HR Institute for University-Industry Cooperation, Keimyung University, 1095 Dalgubeol-daero, Dalseo-gu, Daegu 42601, Korea
* Correspondence: ykh0544@kmu.ac.kr

**Abstract:** This study examined the effect of the Protestant work ethic on burnout using a sample of 259 South Korean workers from a manufacturing firm. We also investigated the mediating role of emotional dissonance on this effect and addressed the moderating and moderated mediating roles of negative emotion regulation on the relationship between Protestant work ethic and emotional dissonance. Our empirical results indicated a significant direct negative effect of the Protestant work ethic on burnout, but there was no evidence of an indirect relationship between these. Results also found that negative emotion regulation changed the relationship between Protestant work ethic and emotional dissonance. In addition, negative emotion regulation changed the mediating role of emotional dissonance in the relationship between Protestant work ethic and burnout. The study is meaningful in that it grasped the importance of value as a major factor in job burnout, and it finally confirmed the antecedents of Koreans' diligence.

**Keywords:** protestant work ethic; burnout; negative emotion regulation; emotional dissonance



## 1. Introduction

According to the IMF's 2019 World Economic Outlook, South Korea ranked 30th in the world with a GDP per capita of USD 31,431 in 2019. However, according to the World Bank, as of 1953, South Korea's GNI per capita was only USD 67, and it was one of the poorest countries in the world. Although the efforts of the state and companies played a large role in this economic growth, the diligence of South Koreans as organizational members also had a significant effect [1]. The job attitude of South Koreans, who do not show tiredness and are devoted to their duties despite very harsh working conditions, can be considered the driving force that led the development of companies and the whole country.

This study focuses on the role of the Protestant work ethic (PWE) on job burnout. This is because existing research on the preceding factors of job burnout mainly studied organizational and individual characteristics, but they also attempted to identify the sources of job burnout by focusing on the characteristics of organizations/jobs and individuals as well as other environmental factors. As representative factors of job burnout, the personal factors include positive psychological capital [2], emotional intelligence [3], self-efficacy [4], locus of control, and emotional stability [5]. Job-related factors include role conflict [6], job overload [7], job performance [8], and job demand–job resources [2,9,10]. Prior studies did not pay attention to the society surrounding the organization, more specifically, the dominant values of the society as antecedents of job burnout. Inglehart [11] argued that through the theory of value change, members and groups belonging to a society are greatly influenced by the dominant values of that society. From this perspective, this study believes

the antecedents of job burnout at the organizational and individual levels remain to be uncovered. This perspective is the starting point of this study.

That is, while previous research has only focused on organizational and individual characteristics as antecedents of burnout, this study made a novel contribution by attempting to find the antecedents of burnout among social values. Normative work values based on religious values such as Protestantism, which appeared as the religious mainstream in South Korean society in the late 20th century, are drawing attention as societal values in addition to individual and organizational characteristics. South Koreans work hard without showing tiredness because they believe that it is worth it to work hard for their jobs, which is a value found in Protestantism. Protestantism and Catholicism were brought from Europe to East Asia in the 19th century, and they are now established as representative religions in South Korea [12]. In accordance with a Gallup Korea survey on religious status in 2015, 21% of South Koreans are Protestant and 7% are Catholic, meaning that 3 of 10 South Koreans are Christians.

Since a PWE applies Christian values to the workplace and emphasizes diligence, abstinence, and success [13], psychological hardship at the workplace is considered as something to be overcome. In other words, Christians are supposed to endure and overcome physical or psychological pain. On the basis of this view, this study sought to confirm the effects of PWE on job burnout. At this point in time, it is important to examine the relationship between PWE and burnout, as South Korea's PWE is now becoming the country's dominant value. The characteristics of PWE may thus be able to explain the exhaustion of South Koreans.

To further develop this discussion, this study employed the value–attitude–behavior (VAB) model of Homer and Kahle [14] and the affective event theory of Weiss and Cropanzano [15] as its theoretical foundation. The VAB model, which denotes that values form the attitudes and behaviors of humans, is considered a cognitive hierarchy model that recognizes the hierarchical influential relationships between values, attitudes, and behavior. Values are a uniform standard of human behavior and an ideology or principle that a person considers as right. Values pursued by individuals are generally in the realm of the values of the organization or society to which they belong. That is, societal and organizational values have an important effect on the attitudes and behaviors of their members [16,17]. This applies to this analysis, which assumes that the 'value' of PWE affects emotional disharmony and job burnout, which are the attitudes and behaviors referred to here. In this study, to explain the role of emotional dissonance and negative emotional regulation as factors that mediate or change the relationship between PWE and job burnout, we also applied Weiss and Cropanzano's [15] affective event theory, which considers that the emotional reactions generated by various events that are experienced at the workplace affect job attitude. Although it was suggested that we use the VAB model as the entire basis of this study, the roles of emotions such as emotional dissonance and negative emotional regulation in the process of PWE affecting job burnout can be better explained by the affective event theory.

Accordingly, this study tried to grasp the influence of religious values, which underlie the human mind, on burnout. By introducing negative emotional control and emotional dissonance, the role of emotion was tested in the relationship between PWE and burnout. This study is of great significance as it indicates how the new major value of the Korean society, PWE, is related to burnout, which is related to the development of the Korean industry, and what role emotion plays in the process.

This paper is composed of introduction, literature review and hypothesis development, methodology, results, and limitation sections. The introduction shows the purpose of the study, the motive for doing the research, and in what ways this paper is valuable. In the literature review and hypothesis development section, hypotheses are derived on the basis of previous studies, and these hypotheses are verified in the methodology and results sections. The discussion and limitation sections focus on the implications of this study.

## 2. Literature Review and Hypothesis Development

### 2.1. PWE and Burnout

Job burnout refers to a state of emotional burnout due to prolonged exposure to psychologically or physically difficult situations [18]. In other words, it is the negative mental state that emerges when highly tense interpersonal relationships are experienced for a long time [19]. Maslach, Schaufeli, & Leiter [20] suggested three subfactors pertaining to emotional burnout—emotional exhaustion, cynicism, and decreased job efficacy—although the construct has not yet been fully defined.

Job burnout negatively affects organizations and individuals, such as increased turnover intention, negative work attitude, and decreased work performance [21]. In addition, it increases the degree of objection to interpersonal relationships and causes difficulties in everyday life [22]. In some cases, it may lead to turnover and a change in jobs [23]. In a study on South Korean employees, Hyun and Hor [24] found that the job burnout of a travel agency's staff decreased job embeddedness. Consequently, job burnout reduces organizational members' passion and devotion to their jobs and eventually leads to poorer performance.

Factors related to jobs most frequently emerge as factors preceding job burnout [25,26]. Typical cases of job overload that influence job burnout include jobs with deadlines [27], lack of practical work performance [28], and insufficient job capacity [29]. Job overload accelerates emotional exhaustion [27], and the effect of role stress is positively related to burnout. Furthermore, vague roles in an organization cause role conflict or role ambiguity, resulting in job burnout. While job overload and role stress are organizational factors that can cause job burnout, emotional labor is a psychological factor. Emotional labor steadily influences surface acting, wherein one hides one's emotions, thus increasing job burnout. Specifically, the effect of surface acting has a positive relationship with burnout and cynicism, among other subfactors [30,31]. In contrast, deep acting, wherein one honestly expresses one's emotions, decreases job burnout [32].

A work ethic is formed through various channels, including religion, which has a great influence. Since religion provides a way of life to be observed as a human being, an individual who believes in a specific religion has no choice but to have a work ethic in the workplace that realizes the values of that religion [33]. In other words, religion forms the basis of the work ethic [34]. The roots of PWE are the Protestantism religion, and PWE is expressed by the value of work in the lives of individuals [35,36]. This concept was initially proposed by Weber [37], who considered work an essential element and viewed working hard for success positively and rest negatively [35,38]. Moreover, Weber thought that the positive view of PWE on the accumulation of wealth led to the development of capitalism in the West. Thus, the concept of PWE can be clarified through its components, namely, hard work, asceticism, and a negative view of leisure, which are common components of success. In addition, internal motives are important elements.

PWE was first presented by Weber [37], who argued that PWE acted as a tool to justify worldly calling to Europeans and to be the ideological basis of modern capitalism. In other words, PWE considered hard work and the wealth gained as a result of that effort should be fully acknowledged. For this reason, PWE has an influence on the job attitude of individuals in an organization, especially the attitude to do something hard without stopping [39]. Therefore, people with high PWE have an attitude of commitment to achieve something [40–42].

As discussed, PWE views hard work positively. It emphasizes considering one's job as a vocation and working hard with an attitude of devotion. This makes it possible to work with passion, even if the work is not favorable. The expectation of PWE is that the current hardships will create something positive in the distant future. People with a high PWE demonstrate diligence [35,38]. Poulton and Ng [43] targeted 68 New Zealand college students to find that students with high levels of PWE worked harder without a break for two weeks. Smrt and Karau [44], in an experimental study of 74 college students, found that students with high PWE generated more ideas than students without PWE. This is because

a PWE frowns upon entertainment activities that do not contribute to the acquisition of economic profit [45]. In other words, individuals with a high PWE do not become stressed, even when facing challenges, because they have higher intrinsic motivation for completing the job [38]. This enables a more devoted effort with less fatigue. Furthermore, those with a high PWE can complete their work on their own without being forced. On the basis of the discussion thus far, we hypothesize that PWE may reduce job burnout.

**Hypothesis 1.** *A PWE is negatively associated with burnout.*

### 2.2. Emotional Dissonance as a Mediator

Emotional dissonance is a state in which the emotional expression of organizational members conforms to the organization's rules for emotional expression but is in conflict with their own internal feelings [46–51]. When dissonance is noticed, employees control their feelings to reconcile emotional expression with the rules for expression. This situation ultimately causes psychological discomfort [52,53]. This study assumes emotional dissonance as a variable that mediates the relationship between PWE and burnout. That is, a person with a high PWE has lower emotional dissonance, which consequently lowers the level of burnout.

Numerous studies have suggested organizational and individual characteristics and task nature, such as emotional labor, as precedence factors of emotional dissonance [46,54,55]. Kwak, McNeeley, & Kim [56] found in a study on Korean police officers that role conflict has a significant effect on emotional exhaustion and cynicism through emotional dissonance. However, research is lacking on religious values that may have a fundamental influence on people's emotions and behaviors. This study focuses on PWE as a religious value that is core to a Western religion and is also representative of Koreans. Previous studies identified PWE as a construct emphasizing hard work, internal motivation, asceticism, and an anti-leisure tendency [57]. McHoskey [36] noted four subfactors as the core values of PWE: success, asceticism, hard work, and anti-leisure tendency. In this view, PWE expects its members to be mature enough to tackle their given jobs in calm and diligent ways. Organizational members may try to force their own emotions to become positive in accordance with the PWE pursued by the individual. Thus, individuals try to embrace attitudes that are deemed desirable by the organization. Rather than emphasizing their own emotions, they attempt to fit their emotions to those stipulated by the organization. As a result, PWE may work as a mechanism to reduce the emotional dissonance of an individual in the organization. This decreased emotional dissonance reduces job burnout.

Thus far, many studies have confirmed that emotional dissonance has a negative effect on the psychological and physiological wellness of workers, such as emotional exhaustion and job burnout [58,59]. Emotional dissonance gives rise to emotional depletion and exhaustion [48], psychological exhaustion [31,60], and decreased job satisfaction [61], thus leading to job burnout.

The mediating role of emotional dissonance has also been confirmed by previous studies. Kwak, McNeeley, & Kim [56] confirmed that emotional dissonance plays a mediating role in the process of role stressors, such as role conflict and ambiguity in affecting burnout. Van Dijk and Brown [51] identified the mediating role of emotional dissonance, such as for mediating emotional labor and emotional exhaustion. Ko and Lee [62] also found that emotional dissonance had a mediating effect on the relationship between emotional labor and burnout among Korean clinical nurses. Accordingly, on the basis of the aforementioned VAB model and affective event theory, we intend to establish a hypothesis that the value of PWE affects the attitude of emotional dissonance, and the affected dissonance affects burnout. We propose a previously unexplored hypothesis that PWE may have an indirect effect on burnout through emotional dissonance.

**Hypothesis 2.** *Emotional dissonance mediates the relationship between PWE and burnout.*

*2.3. Negative Emotion Regulation as a Moderator*

The regulation of emotion, especially negative emotion, is a conscious or unconscious procedure that increases pleasant emotions and minimizes unpleasant emotions [63]. Most previous studies focused on the suppression of negative emotions [64,65]. Individuals who can regulate negative emotions tend to have good relationships with others in the organization [66,67]. Those who control negative emotions are more likely to become socially and emotionally competent and may establish healthy interactions with others in the organization [68]. Good control over negative emotions can prevent unnecessary conflict with others in the organization and culminate in protecting oneself [69]. It thus has an important role in the workplace, as confirmed Sung, Choi, & Chun [70], who found that the better people can control negative emotions, the higher is their job satisfaction.

Although the relationship between PWE and emotional dissonance can be explained through the VAB model, overlooking the relationship with other elements when considering the relationship between PWE as a value and emotional dissonance as an emotion oversimplifies these two concepts. Even though a person is religiously mature, religious values often become meaningless if the person does not have good innate property. Therefore, a pastor may lose control over their emotions, becoming angry in difficult situations, although this is not usually the case. People with a high capacity to regulate emotion control and express their emotions try to embrace a more desirable attitude, which is expressed in a way that reduces emotional dissonance. Thus, those with good control over negative emotions may reduce emotional dissonance with PWE [71]. However, regardless of their level of PWE, those with poor control over negative emotions may not be able to reduce emotional dissonance. In other words, PWE may have a negative influence on emotional dissonance when the level of regulation of negative emotion is high. Specifically, the higher the level of negative emotion regulation, the greater the negative effect of PWE on emotional dissonance.

**Hypothesis 3.** *Negative emotion regulation moderates the relationship between PWE and emotional dissonance.*

*2.4. The Moderated Mediation Effect of Negative Emotion Regulation*

This study combines Hypotheses 2 and 3, respectively stating that emotional dissonance mediates PWE and emotional dissonance, and that negative emotion regulation controls PWE and emotional dissonance. The purpose was to hypothesize mediated moderation [72]; in other words, the interaction effect between PWE and negative emotion regulation is conveyed by emotional dissonance to affect burnout. The mediated moderation effect means that the level of mediation effect varies according to the moderator level. The regulation of negative emotions is an important factor that enables PWE performance. When negative emotion regulation is high, the influence of PWE on emotional dissonance and burnout eventually expands. On the basis of this viewpoint, this study proposes the adoption of Hypothesis 4, which assumes that the indirect effect through emotional dissonance may hold only when negative emotion regulation is high.

**Hypothesis 4.** *The indirect effect of PWE on burnout through emotional dissonance holds only for higher levels of negative emotion regulation.*

## 3. Methodology

*3.1. Participants and Procedure*

The participants in this study were employees working in companies located in South Korea. Data were collected through both online and offline surveys. Participants that were recruited online anonymously completed an online questionnaire. The survey was not conducted by selecting a specific company; workers from South Korean companies were randomly asked to complete the questionnaire. We collected data from 275 employees

working for private companies. Of these, 16 questionnaires were eliminated for incompleteness. In total, 157 (60.5%) participants were male and 102 (39.5%) were female. The ages of the participants were relatively equally distributed: 32.0% were aged 20–29 years, 33.6% were 30–39 years, and 23.2% were 40–49 years of age. The job tenure varied. The majority had worked for less than 6 years (39.1%), followed by 11 to 15 years (19.4%), 6 to 10 years (17.8%), more than 21 years (13.8%), and 15 to 20 years (9.9%). Religion was 25% Protestant, 8% Catholic, and 7% Buddhist, and the rest were nonreligious.

The English version of the questionnaire was translated into Korean, then back-translated into English. Three bilingual professionals were involved in the English–Korean translation. As the study was conducted in Korean, scale items not available in Korean were translated using a standard translation to back-translation procedure [73].

### 3.2. Measures

#### 3.2.1. Protestant Work Ethic

Protestant work ethic (PWE) was measured using the PWE scale by Mirels and Garrett [74]. The scale has 19 items that were responded to using a seven-point scale where 1 = "strongly disagree" and 7 = "strongly agree." Sample items include, "People should have more leisure time to spend relaxing" (reverse scored) and "There are few satisfactions equal to the realization that one has done his best at a job". Responses to the items were summed, and higher scores indicate higher PWE.

#### 3.2.2. Emotional Dissonance

To assess emotional dissonance (ED), we used the three-item measure by Morris and Feldman [75] and the two-item measure by Brotheridge and Lee [76]. Items were responded to using a seven-point rating scale ranging from 1 = "strongly disagree" to 7 = "strongly agree". Sample items include, "When I work with customers/clients, the way I act and speak often does not match what I really feel" and "Pretend to have emotions that you do not really feel". Responses to the items are summed, and higher scores indicate higher ED.

#### 3.2.3. Burnout

The 16-item Maslach Burnout Inventory–General Survey (MBI-GS) [77] was used to assess occupational burnout. The items were responded to using a seven-point scale ranging from 1 = "strongly disagree" to 7 = "strongly agree". Sample items include, "I feel emotionally drained by my work" and "I have become less enthusiastic about my work". Responses to the items are summed, and higher scores indicate higher levels of burnout. The internal consistency reliability of the measure was 0.92 in the current study.

#### 3.2.4. Negative Emotion Regulation

Negative emotion regulation (NER) was measured using 12 items from the 30-item Negative Mood Regulation Scale (NMR) [78] that were best suited for the purposes of the current study. A sample item is, "I can usually find a way to cheer myself up". Items were responded to using a seven-point scale where 1 = "strongly disagree" and 7 = "strongly agree". Responses to the items are summed, and higher scores indicate higher NER. Cronbach's alpha in the current study was 0.84.

### 3.3. Data Analysis

Descriptive statistics were calculated for the variables including means, standard deviations, and correlations. Confirmatory factor analysis (CFA) was used to determine the reliability and validity of the scales. For model verification, SPSS 25.0 PROCESS macro was implemented on the basis of the conditional indirect effect verification by Hayes [79]. First, we used the analysis method proposed by Baron and Kenny [80] to verify the direct relationship, modulating effect, and mediating effect between major variables. Second, we used the PROCESS macro model 7 to verify the moderation effect of NER on the effect of PWE on ED. We then investigated the confidence intervals of the conditional indirect effect

for high, middle, and low moderator variable levels to verify the moderated mediation effect of ED.

## 4. Results

### 4.1. Preliminary Analyses

First, a CFA was conducted for each individual construct, and second, for the overall measurement model in which all the major latent constructs were correlated with each other. As recommended by Hair et al. [81], a combination of model fit indices was employed to assess the fit. The normed chi-squared ($\chi^2$/df), standardized root-mean-square residual (SRMR), and root-mean-square error of approximation (RMSEA) were applied as absolute fit indices. In addition, the comparative fit index (CFI) and Tucker–Lewis index (TLI) were used as incremental fit indices.

Although the chi-squared test was statistically significant, the other model fit indices were sufficiently acceptable (TLI = 0.91, CFI = 0.92, SRMR = 0.06, RMSEA = 0.06 (90% confidence interval 0.37~0.83)). The results of the model fit indices indicated that the final measurement model fit the data reasonably well. Table 1 reports the descriptive statistics, correlations, and average variance extracted (AVE) for the key study variables. All AVEs were greater than 0.50 (AVE ranged from 0.53–0.71), thus demonstrating convergent validity [82]. Discriminant validity was tested using the Fornell and Larcker [83] approach, wherein the AVEs are compared to the shared variances (i.e., construct correlations squared). This procedure resulted in all AVEs being greater than the shared variances, indicating discriminant validity. In addition, the reliability coefficients of the measures ranged from 0.76 to 0.92, indicating they were generally reliable. After validity analysis and reliability analysis, the arithmetic mean of the items belonging to each variable was obtained and used for correlation analysis and hypothesis verification.

**Table 1.** Descriptive statistics, correlations, and AVEs for key study variables.

| Variable | Mean | SD | 1 | 2 | 3 | 4 |
|---|---|---|---|---|---|---|
| 1. PWE | 30.94 | 0.67 | 0.75 (0.78) | | | |
| 2. ED | 40.85 | 10.03 | 0.14 | 0.77 (0.76) | | |
| 3. Burnout | 30.44 | 0.98 | −0.17 * | | 0.84 (0.92) | |
| 4. NER | 40.62 | 0.83 | 0.35 ** | | | 0.73 |

* $p < 0.05$; ** $p < 0.01$. Note: $N = 259$. Sub-diagonal entries are latent construct inter-correlations. The first entry on the diagonal is the square root of the AVE, while the second entry in parentheses is the composite reliability score. AVE = average variance extracted; PWE = Protestant work ethic; ED = emotional dissonance; NER = negative emotion regulation.

Because all the variables were measured using the same source in the current study, the effects of common method bias were examined. Harman's single-factor test was conducted with the use of the exploratory factor analysis (EFA) to check for the presence of common method bias. According to Harman's Single Factor Test, if one factor occupies more than 50% of the variance, there is CMB. In this study, the largest variance was only 8.34%, suggesting that common method bias is unlikely to be a concern in this study.

### 4.2. Hypothesis Testing

First, Hayes' PROCESS macro model 4 analysis was performed to investigate the influence of PWE on burnout and the mediating role of ED. Gender, age, job tenure, and firm size were control variables. The results are displayed in Table 2. PWE had a significant negative effect on burnout, β = −0.33, $p < 0.01$. There was no mediating effect of ED on the relationship between PWE and burnout. Therefore, while Hypothesis 1 was supported, Hypothesis 2 was not supported.

**Table 2.** The relationships among PWE, ED, and burnout in the mediation model.

| Variable | ED | | Burnout | |
|---|---|---|---|---|
| | β | SE | β | SE |
| Control variables | | | | |
| Gender | 0.19 | 0.14 | 0.14 | 0.11 |
| Age | 0.01 | 0.13 | 0.01 | 0.11 |
| Job tenure | −0.07 | 0.09 | −0.01 | 0.08 |
| Firm size | 0.04 | 0.06 | −0.11 * | 0.06 |
| Independent variable | | | | |
| PWE | 0.11 | 0.10 | −0.33 ** | 0.09 |
| Mediator | | | | |
| ED | | | 0.38 ** | 0.05 |
| *F* | 10.27 | | 120.01 ** | |
| $R^2$ | 0.02 | | 0.23 | |
| Indirect effect of ED on the relationship between PWE and burnout | | | | |
| | β | *SE* | LLCI | UCLI |
| ED | 0.04 | 0.04 | −0.04 | 0.13 |

* $p < 0.05$; ** $p < 0.01$. Note: *N* = 259. PWE = Protestant work ethic; ED = emotional dissonance; LLCI = lower level of 95% confidence interval; ULCI = upper level of 95% confidence interval.

Next, in order to investigate the moderating effect of NER and the moderated mediating effect, we performed Hayes' PROCESS macro model 7. The results are shown in Table 3. NER moderated the relationship between PWE and burnout, β = −0.33, *p* < 0.01. Hypothesis 3 was not supported. The index of moderated mediation was significant, β = 0.10, 95% CI [0.02, 0.20], indicating that the indirect effect of ED was moderated. See Table 4. When ED was at a lower level and at a moderate level, the conditional indirect effect of PWE on burnout via ED was not significant. In contrast, when ED was at a higher level, the effect was significant, β = 0.15, 95% CI [0.04, 0.27]. In sum, the results supported H4, indicating that NER was a moderator of the pathway from PWE through ED to burnout. It is meaningful to verify the conditional indirect effect in that it was found that ED mediated the relationship between PWE and burnout only when NER was high (+1 SD), even though a significant mediating effect of ED was not found (see Figure 1).

**Table 3.** The moderated mediation model of PWE, ED, and NER predicting burnout.

| Variable | ED | | Burnout | |
|---|---|---|---|---|
| | β | SE | β | SE |
| Control variables | | | | |
| Gender | 0.13 | 0.12 | 0.15 | 0.11 |
| Age | 0.01 | 0.07 | 0.01 | 0.11 |
| Job tenure | −0.04 | 0.05 | −0.04 | 0.05 |
| Firm size | 0.02 | 0.03 | −0.02 | 0.03 |
| Independent variable | | | | |
| PWE | −0.22 * | 0.10 | −0.29 ** | 0.08 |
| Moderator | | | | |
| NER | −0.32 ** | 0.10 | −0.38 ** | 0.05 |
| Independent variable × moderator | | | | |
| PWE × NER | 0.28 ** | 0.09 | | |
| *F* | 70.61 ** | | 80.62 ** | |
| $R^2$ | 0.15 | | 0.23 | |

* $p < 0.05$; ** $p < 0.01$. Note: *N* = 259. PWE = Protestant work ethic; ED = emotional dissonance; NER = negative emotion regulation.

**Table 4.** The moderated mediation effect of NER.

| NER | β | SE | LLCI | UCLI |
|---|---|---|---|---|
| Low (−1 SD) | −0.02 | 0.04 | −0.12 | 0.06 |
| Moderate (mean) | 0.05 | 0.03 | −0.01 | 0.13 |
| High (+1 SD) | 0.15 | 0.06 | 0.04 | 0.27 |
| Index of moderated mediation | | | | |
| Variable | β | *SE* | LLCI | UCLI |
| NER | 0.10 | 0.04 | 0.02 | 0.20 |

* $p < 0.05$; ** $p < 0.01$. Note: $N = 259$. NER = negative emotion regulation; LLCI = lower level of 95% confidence interval, ULCI = upper level of 95% confidence interval.

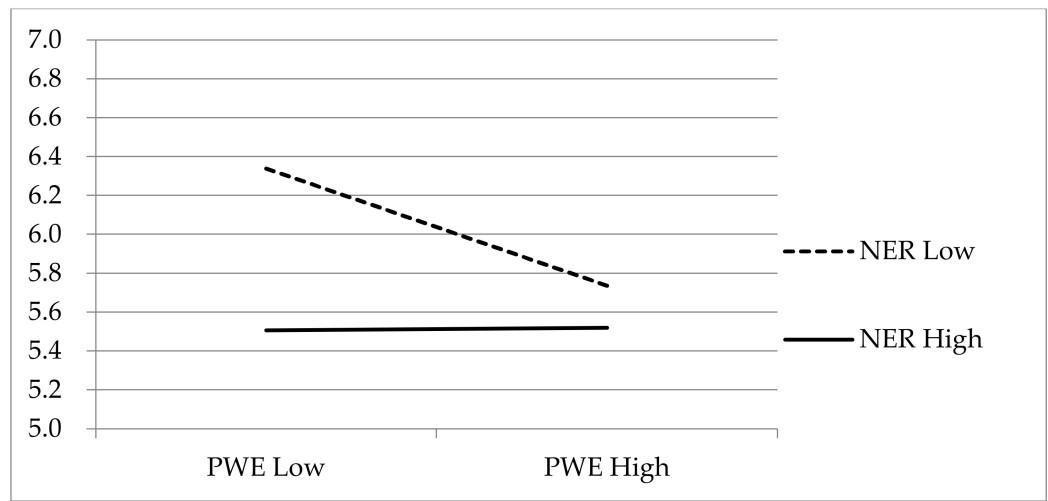

**Figure 1.** The moderating effect of NER on the relationship between PWE and emotional dissonance.

## 5. Discussion

In this study, we examined the relationship between the Protestant work ethic (PWE) and occupational burnout, the mediating role of emotional dissonance in the relationship between PWE and burnout, and the moderating role of negative emotion regulation in the relationship between PWE and burnout mediated by emotional dissonance. The results showed PWE was negatively associated with burnout. Negative emotion regulation moderated the relationship between PWE and emotional dissonance as well as the mediating effect of emotional dissonance between PWE and burnout. No mediating effect was found for the relationship between PWE and burnout.

These findings can be considered to be of great importance, in that they provide a possible explanation for how PWE, which is emerging as a major cultural value in South Korea, has the potential to prevent burnout. In fact, until the late 20th century, the attitudes and behaviors of individuals in South Korean society were greatly influenced by Confucian values. In the 21st century, the Protestant work ethic has been replacing the Confucian work ethic. In other words, as a factor influencing feelings and attitudes towards one's job, such as occupational burnout, the influence of the traditional Confucian work ethic has decreased, and the Western influence of the Protestant work ethic has increased. No prior study has investigated the influence of PWE on burnout at the individual level of analysis in South Korea. The results of the present study showed that PWE was negatively associated with burnout, which is consistent with Organ and Hui's [39] study, which suggested organization citizenship behavior was associated with aspects of PWE that value hard work and the attitude that no time is available for relaxation. Mirles and Garrett [74] considered as key elements of PWE the notion that one must give up pleasure and work hard for success, or that the meaning of life can only be understood by experiencing

hardship. The Bible has a specific call to action that "if any would not work, neither should he eat" (2 Thessalonians 3:10), which provides followers with a general guideline. According to PWE, the present hardship is perceived as a rite of passage one must go through for a bright future. Therefore, the challenges experienced in the organization must be endured and overcome; that is, current suffering is viewed as something to be endured. Thus, one can accept the present difficulties, which counteracts feeling tired or burnout. The results in the present study provide new knowledge on the association between PWE and burnout in South Korean workers, which may be due to PWE as a value emerging in South Koreans and as an important aspect of organizational culture in South Korea.

We contribute to the literature on emotional dissonance by introducing negative emotion regulation as a psychological variable that interacts with PWE to influence emotional dissonance. The results revealed negative emotion regulation strengthened the effect of PWE on emotional dissonance as a mediator, showing that PWE can play an even greater role on burnout when negative emotion regulation is low. Therefore, being able to regulate negative emotions can be viewed as a critical condition to increase the negative effect of PWE on burnout. In contrast, there was no mediating effect of emotional dissonance on the relationship between PWE and burnout. It is possible that having a strong PWE would not result in emotional dissonance because there would be no conflict between one's experienced emotions and those that are to be displayed to conform at the workplace.

This study is significant in that it revealed that the mediating effect of emotional dissonance was confirmed when negative emotional regulation was added. More specifically, the mediating role of emotional dissonance in the relationship between PWE and burnout was verified for people with high negative emotion regulation, the strong ability to manage negative emotions. This implies that the relationship between PWE and burnout can be further explained through various third variables. In other words, we can see that the effect of PWE on burnout can be changed by its relationship with other variables, and the effect is complicated.

From a practical point of view, we suggest some implications based on our findings. The results of the current study show that PWE helps employees to think positively about their hard work. In recruiting and selecting workers, organizations might consider characteristics associated with the Protestant work ethic but not particular to Protestantism, such as viewing hard work as good and work as having dignity and being high achievers who are committed to the organization. In the hiring and selection process, human resource personnel could use a measure to assess such characteristics in addition to a job aptitude test. Such an assessment would be particularly useful in organizations that have jobs that require high emotional labor or high intensity of work.

Given the complex roles of emotional dissonance and negative emotion regulation in the relationship between PWE and burnout, organizations should make various efforts to keep the emotions of the members positive in order to prevent burnout. It is necessary to provide psychological care and establish a working environment to prevent emotional dissonance. Given values such as PWE are difficult to change by the organization, the culture of the organization should be one that encourages positive emotions among its members so that the need to regulate negative emotions would not be because of negative emotions created at the workplace.

## 6. Limitations

Due to several limitations of the current study, interpretation of the findings should be done with caution. First, all the variables in the current study were measured on a seven-point Likert scale. Therefore, a common method bias may occur. The results of the study could be expanded through different methods of data collection, including qualitative methods such as in-depth interviews. Mixed methods could be employed to increase the validity of the results.

Second, the study examined psychological constructs that were not easy to observe. Although the reliability and validity of the collected data were reviewed, these results do

not guarantee the adequacy of the constructs and measurements, given that all measures in this study were self-reported. Moreover, all measures of the study variables were developed outside South Korea. Thus, the conceptual homogeneity of constructs cannot be guaranteed, making it difficult to conclude that the validity of the data was fully established.

Third, there is a considerable conceptual distance between the religious value of the Protestant work ethic and the emotions and attitudes of individuals in the workplace, such as job burnout, and many other variables may play a role in the relationship between these variables. Nevertheless, this study has a limitation in that it was only able to consider two of the many complex and various realistic situations that can affect burnout. Therefore, future studies need to examine the effects of other moderators and mediators not included in this study.

Fourth, this study may have limitations in that it is a sample from a country with a high proportion of Protestants, even though South Korea is a religiously diverse country (Buddhism, Christianity, and nonreligious). In order to increase external validity, research with samples from diverse countries will be necessary. In future studies, the role of PWE on burnout should be verified in Western culture, which can be said to be the origin of PWE.

Lastly, future research will need to investigate organizational performance. Research on how values at the individual level affect productivity at the organizational level is lacking. It can be shown that an individual's organizational behavior, which is related to the productivity of the organization, can be influenced by the values of the society. Although the organization puts a lot of effort into introducing various personnel systems and improving the organizational culture in order to change the organizational behavior of its members, it is an individual's values or social values that primarily affects the individual. In other words, it is necessary to study the effect of specific values on organizational behaviors that are important in a specific society or occupation, such as obedience to orders of superiors in military organizations.

**Author Contributions:** This paper was written by S.C., Y.K., and K.Y. All authors have read and agreed to the published version of the manuscript.

**Funding:** This research received no external funding.

**Institutional Review Board Statement:** Not applicable.

**Informed Consent Statement:** Informed consent was obtained from all subjects involved in the study.

**Data Availability Statement:** Data presented in this manuscript are available upon request from the corresponding author.

**Conflicts of Interest:** The authors declare no conflict of interest.

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
