# Peer review of "Effect of a Protestant Work Ethic on Burnout: Mediating Effect of Emotional Dissonance and Moderated Mediating Effect of Negative Emotion Regulation"

_sustainability, doi:10.3390/su13115909_

Round 1

Reviewer 1 Report

The article is interesting, and the topic is of practical importance. My comments and recommendations are presented in what follows:

  1. Lines 21-24: Why is this clarification important? This must be explained in the abstract. We don't care about clarifications of matters that are not important in any way. What are the implications of the results? In what sense is the study important?
  2. Lines 28-32: Some references are needed - they could be provided by the National Institute of Statistics, by the World Bank, etc, but you need to indicate sources of the figures.
  3. Lines 37-39: this sentence does not seem related to the information above. 
  4. Lines 40-46: the information is correct, but it needs to be organized around a clearly specified research question. 
  5. Lines 58-59: the authors say that "However, research on these topics is lacking". My question is: is lacking completely, or there is something that has been done before?
  6. Line 82: the authors say that they want to assess the effects of PWE on burnout, and they indeed conduct a path analysis. However, they work with survey data, which is not enough to prove causality. Maybe the author can reconsider this sentence.
  7. Figure 1: an e is missing at the end of Dissonance
  8. Lines 325-326: the authors write that they conduct CFA and used latent constructs, which is good. However, in section 3.2. they claim that they scored their questionnaires by adding up the respondents' answers. This needs clarifications.
  9. Lines 332-333: please also report SRMR, and confidence intervals for the fit measures.
  10. Lines 350-356: I appreciate that you conducted Hartman's test for common method bias. However, you need to report how much variance captures that unique factor: although I understand the argument related to the poor performance of the model, the test requires at least 50% variance explained to confirm the bias. Therefore you need to conduct exploratory factor analysis and report the variance extracted, and not confirmatory factor analysis. 
  11. What are the columns in Table 2 and Table 3?
  12. The manner in which the authors report the results is unusual, and hard to comprehend. Maybe they reconsider their approach and try to report the results in the standard way SEM is reported. You need to report direct, indirect, and total effects in the same table, so the decomposition of the total effect becomes clear. Effect sizes must be reported in a separate table, and discussed in line with the theoretical requirements.  The same applies to the mediated moderation effect, it has to be reported in a separate table and discussed accordingly. 
  13. Section 5 - Discussion appears twice.

Author Response

Response to Reviewer 1 Comments

Point 1: Lines 21-24: Why is this clarification important? This must be explained in the abstract. We don't care about clarifications of matters that are not important in any way. What are the implications of the results? In what sense is the study important?

Response 1: We erased the existing text and put in a new one. The new article is as follows.

“The significance of this study is that it shows changes in the dominant values that affect the organizational behavior of Koreans, and reveals how the relationship can be changed by individ-ual emotional factors.”

Point 2: Lines 28-32: Some references are needed - they could be provided by the National Institute of Statistics, by the World Bank, etc, but you need to indicate sources of the figures.

Response 2: We erased the existing text and put in a new one. The new article is as follows.

“According to the IMF's 2019 World Economic Outlook, Korea ranked 30th in the world with a GDP per capita of $31,431 in 2019. However, according to the World Bank, as of 1953, Korea's GNI per capita was only $67, and it was one of the poorest countries in the world.”

Point 3: Lines 37-39: this sentence does not seem related to the information above.

Response 3: We deleted the sentence.

Point 4: Lines 40-46: the information is correct, but it needs to be organized around a clearly specified research question. 

Response 4: We deleted the sentence.

Point 5: Lines 58-59: the authors say that "However, research on these topics is lacking". My question is: is lacking completely, or there is something that has been done before?

Response 5: We erased the existing text and put in a new one. The new article is as follows.

“However, there is very little on these topics.”

Point 6: Line 82: the authors say that they want to assess the effects of PWE on burnout, and they indeed conduct a path analysis. However, they work with survey data, which is not enough to prove causality. Maybe the author can reconsider this sentence.

Response 6: We erased the existing text and put in a new one. The new article is as follows.

“it is important to examine the relationship between PWE and burnout”

Point 7: Figure 1: an e is missing at the end of Dissonance

Response 7: We added the ‘e’ at the end of Dissonance

Point 8: Lines 325-326: the authors write that they conduct CFA and used latent constructs, which is good. However, in section 3.2. they claim that they scored their questionnaires by adding up the respondents' answers. This needs clarifications.

Response 8: It is not the summation of the answers, but the average of the items.

Point 9: Lines 332-333: please also report SRMR, and confidence intervals for the fit measures.

Response 9: We reflected the reviewer's requirements.

Point 10: Lines 350-356: I appreciate that you conducted Hartman's test for common method bias. However, you need to report how much variance captures that unique factor: although I understand the argument related to the poor performance of the model, the test requires at least 50% variance explained to confirm the bias. Therefore you need to conduct exploratory factor analysis and report the variance extracted, and not confirmatory factor analysis. 

Response 10: We erased the existing text and put in a new one. The new article is as follows.

“Harman’s single-factor test was conducted with the use of the exploratory factor analysis (EFA) to check for the presence of common method bias. According to Harman's Single Factor Test, if one factor occupies more than 50% of the variance, there is CMB. In this study, the largest variance was only 8.34%, suggesting that common method bias is un-likely to be a concern in this study.”

Point 11: What are the columns in Table 2 and Table 3?

Response 11: We deleted the columns.

Point 12: The manner in which the authors report the results is unusual, and hard to comprehend. Maybe they reconsider their approach and try to report the results in the standard way SEM is reported. You need to report direct, indirect, and total effects in the same table, so the decomposition of the total effect becomes clear. Effect sizes must be reported in a separate table, and discussed in line with the theoretical requirements.  The same applies to the mediated moderation effect, it has to be reported in a separate table and discussed accordingly. 

Response 12: We are not exactly doing SEM. We did CFA for validation. So we stayed in the measurement model. Since it was not an SEM, the results cannot be classified into direct, indirect, and total effects. And the mediated moderation effect was organized in a separate table.

Point 13: Section 5 - Discussion appears twice.

Response 13: We fixed it.

Reviewer 2 Report

The paper aims to provide an interesting analysis on Protestant Work Ethic (PWE) as a religious value that is core to a Western religion and it is also typic of Koreans. By using a sample of 259 South Korean worker the study examines the effect of PWE on burnout. Data were collected through both online and offline surveys. The analysis gives an original contribution in order to find the antecedents of burnout among social values.

The article has potential to be published but the authors must operate some revisions in order to give consistency to the manuscript.

  • Authors have to present the structure of the paper.
  • Authors have to Improve the literature review;
  • 1 must be improved to better and in a more fascinating way explain the hypothesis.
  • Additional graphic representations should be presented in order to better explain the hypothesis content.
  • There are two Par. 5 Discussion (400 and 469).

The shift and changes from Confucianism to the Protestant religion should be stressed.  Questions may arise on the fact that somehow Confucianism can still have an influence on the behavior of workers. What was the vision of the work ethic for Confucianism?

Author Response

Response to Reviewer 2 Comments

Point 1: Authors have to present the structure of the paper.

Response 1: We have added the following.

“This paper is composed of Introduction, literature review and hypothesis develop-ment, methodology, results, and limitation. Introduction showed the purpose of the study, the motive for doing the research, and in what ways this paper is valuable. In literature review and hypothesis development, hypotheses were derived based on previous studies, and hypotheses were verified in methodology and results. The discussion and limitations focused on the implications of this study.”

Point 2: Authors have to improve the literature review

Response 2: We have added the following. Although we tried to add a literature review, there are not many prior studies related to the subject of this study. And in terms of constructing the logic, it is difficult to add a literature review. I hope you understand this point

“PWE was first presented by Weber (1905), who argued that PWE acted as a tool to justify worldly calling to Europeans and to be the ideological basis of modern capitalism. In other words, PWE considered hard work and the wealth gained as a result of that effort should be fully acknowledged. For this reason, PWE has an influence on the job attitude of individuals in an organization, especially the attitude to do something hard without stopping (Organ and Hui, 1995). Therefore, people with high PWE have an attitude of commitment to achieve something (Mudrack, 1997; Kidron, 1978; Randall and Cote, 1991).”

Point 3: 1 must be improved to better and in a more fascinating way explain the hypothesis.

Response 3: I'm not sure exactly what '1' means in the question.

Point 4: Additional graphic representations should be presented in order to better explain the hypothesis content.

Response 4: hypotheses were entered into the research model(figure 1).

Point 5: There are two Par. 5 Discussion (400 and 469).

Response 5: We fixed it.

Point 6: The shift and changes from Confucianism to the Protestant religion should be stressed.  Questions may arise on the fact that somehow Confucianism can still have an influence on the behavior of workers. What was the vision of the work ethic for Confucianism?

Response 6: The influence of Confucian values ​​in S. Korea will continue to diminish. Confucian values ​​are likely not to exist in the 22nd century in Korean society. However, I am in doubt as to whether it is correct to include this part in this paper. Our paper is not about Confucian values, but about PWE. Should I write about the future of Confucian values in the paper?

Round 2

Reviewer 1 Report

The authors revised their paper to a certain extent. However, there are recommendations to which they did not respond. 

Lines 21-23: They read now as "The significance of this study is that it shows changes in the dominant values that affect the organizational behavior of Koreans, and reveals how the relationship can be changed by individual emotional factors". The authors seem to have not understood my question. Their new sentence shows what they did, but they do not provide any explanation of why knowing all this stuff is important in practice. 

Lines 47-48 now read "There is little on this topic". The authors need to expand this part by explaining what that "little" is, what has been done, and in which way their research is different from what exists already. With references, of course.

Point 8 in my previous report: "The authors write that they conduct CFA and used latent constructs, which is good. However, in section 3.2. they claim that they scored their questionnaires by adding up the respondents' answers. This needs clarifications." The authors' answer makes me think that they didn't understand what they need to do. CFA and scoring (either sum of items or average) are two completely different things. This aspect still needs clarification.

In regards to my 12th comment, the authors respond as follows: "We are not exactly doing SEM. We did CFA for validation. So we stayed in the measurement model. Since it was not an SEM, the results cannot be classified into direct, indirect, and total effects. And the mediated moderation effect was organized in a separate table."

This answer makes me think that the authors lack a basic understanding of their own methodology. Figure 1 shows a structural model, in which you hypothesize a mechanism (Emotional Dissonance) that explains how the relationship between PWE and Burnout holds. Tables 2 and 3 show the results of a path model, along with an indirect effect that is not statistically significant (see the last lines of Table 2). Yet, the authors respond that they conducted CFA, not SEM...

Let me say that CFA is nothing but a procedure to test whether a set of data fits a theoretical model - a structural model, in your case - based on the Maximum Likelihood estimation method or other methods that account for non-normality (if the case). Therefore, for the records, yes, you conducted SEM...

The paper needs extensive English language revisions. The authors do not make grammar mistakes, but in some cases, the sentences sound artificial, and the wordiness must be reduced. 

Author Response

Point 1: Lines 21-24: Why is this clarification important? This must be explained in the abstract. We don't care about clarifications of matters that are not important in any way. What are the implications of the results? In what sense is the study important?

Response 1: We erased the existing text and put in a new one. The new article is as follows.

“The significance of this study is that it shows changes in the dominant values that affect the organizational behavior of Koreans, and reveals how the relationship can be changed by individ-ual emotional factors.”

Point 1-1. Lines 21-23: They read now as "The significance of this study is that it shows changes in the dominant values that affect the organizational behavior of Koreans, and reveals how the relationship can be changed by individual emotional factors". The authors seem to have not understood my question. Their new sentence shows what they did, but they do not provide any explanation of why knowing all this stuff is important in practice.

Response 1-1.  I deleted the existing sentence and added the sentence below.

The study is meaningful in that it grasped the importance of value as a major factor in job burn-out, and it finally confirmed the antecedents of Koreans' diligence.

Point 5: Lines 58-59: the authors say that "However, research on these topics is lacking". My question is: is lacking completely, or there is something that has been done before?

Response 5: We erased the existing text and put in a new one. The new article is as follows.

“However, there is very little on these topics.”

Point 5-1: Lines 47-48 now read "There is little on this topic". The authors need to expand this part by explaining what that "little" is, what has been done, and in which way their research is different from what exists already. With references, of course.

Response 5-1. I deleted the sentence, “Although job burnout represents psychological exhaustion and physical fatigue, it is likely influenced by personal beliefs. However, there is very little on these topics. Moreover,” We cannot find any studies dealing with the relationship between personal belief and burnout. However, it is not easy to conclude that there is no such study.

Point 8: Lines 325-326: the authors write that they conduct CFA and used latent constructs, which is good. However, in section 3.2. they claim that they scored their questionnaires by adding up the respondents' answers. This needs clarifications.

Response 8: It is not the summation of the answers, but the average of the items.

Point 8-1. Point 8 in my previous report: "The authors write that they conduct CFA and used latent constructs, which is good. However, in section 3.2. they claim that they scored their questionnaires by adding up the respondents' answers. This needs clarifications." The authors' answer makes me think that they didn't understand what they need to do. CFA and scoring (either sum of items or average) are two completely different things. This aspect still needs clarification.

Response 8-1: We tried to test the validity through CFA, and the hypothesis test was performed by calculating the average of each variable, not the Latent variable. I uploaded a reference paper, so take a look. CFA is performed for validity and regression analysis can be performed for hypothesis testing. It is understood that just because a latent variable is used for CFA does not necessarily mean that an analysis using a latent variable, that is, a structural equation, should be performed when testing the hypothesis. In addition, 7-point Likert scale was used for all variables. I said that each item was measured on a 7-point scale in 3.2, but I didn't add it up. Are you saying that we have to test the hypothesis with a latent variable because we did CFA? CFA was performed to analyze the validity, and the average of each scored variable was obtained, and the relationship between each variable was verified by regression analysis. I think that's a possible way.

Point 12: The manner in which the authors report the results is unusual, and hard to comprehend. Maybe they reconsider their approach and try to report the results in the standard way SEM is reported. You need to report direct, indirect, and total effects in the same table, so the decomposition of the total effect becomes clear. Effect sizes must be reported in a separate table, and discussed in line with the theoretical requirements.  The same applies to the mediated moderation effect, it has to be reported in a separate table and discussed accordingly. 

Response 12: We are not exactly doing SEM. We did CFA for validation. So we stayed in the measurement model. Since it was not an SEM, the results cannot be classified into direct, indirect, and total effects. And the mediated moderation effect was organized in a separate table.

Point 12-1. This answer makes me think that the authors lack a basic understanding of their own methodology. Figure 1 shows a structural model, in which you hypothesize a mechanism (Emotional Dissonance) that explains how the relationship between PWE and Burnout holds. Tables 2 and 3 show the results of a path model, along with an indirect effect that is not statistically significant (see the last lines of Table 2). Yet, the authors respond that they conducted CFA, not SEM...

Let me say that CFA is nothing but a procedure to test whether a set of data fits a theoretical model - a structural model, in your case - based on the Maximum Likelihood estimation method or other methods that account for non-normality (if the case). Therefore, for the records, yes, you conducted SEM...

Response 12-1. If so, I will delete figure 1. I don't know if I'm lacking in understanding, but I don't think figure 1 shows the structural model. CFA was done for validation. The hypothesis test was analyzed using the analysis method of Baron & Kenny (1986) and Hayes' process. These are certainly not SEMs. It may be something I am not sure about, but I have no intention of testing the hypothesis through SEM. The CFA is for validation purposes only. So I deleted Figure 1.
